# Protective Effect of *Eurotium cristatum* Fermented Loose Dark Tea and *Eurotium cristatum* Particle on MAPK and PXR/AhR Signaling Pathways Induced by Electronic Cigarette Exposure in Mice

**DOI:** 10.3390/nu14142843

**Published:** 2022-07-11

**Authors:** Shuai Xu, Yufei Zhou, Lijun Yu, Xiangxiang Huang, Jianan Huang, Kunbo Wang, Zhonghua Liu

**Affiliations:** 1Key Laboratory of Tea Science of Ministry of Education, National Research Center of Engineering Technology for Utilization of Functional Ingredients from Botanicals, College of Horticulture, Hunan Agricultural University, Changsha 410128, China; xushuai199884@163.com (S.X.); zhouyufeittea@stu.hunau.edu.cn (Y.Z.); hxx70345@outlook.com (X.H.); jian7513@hunau.edu.cn (J.H.); wkboo163@163.com (K.W.); 2Hunan Provincial Key Laboratory for Germplasm Innovation and Utilization of Crop, Hunan Agricultural University, Changsha 410128, China

**Keywords:** electronic-cigarette smoke, *Eurotium cristatum*, *Eurotium cristatum* particle, *Eurotium cristatum* fermented loose dark tea, MAPK, PXR, AhR, metabonomics

## Abstract

Electronic-cigarette smoke (eCS) has been shown to cause a degree of oxidative stress and inflammatory damage in lung tissue. The aim of this study was to evaluate the repair mechanism of *Eurotium cristatum* fermented loose dark tea (ECT) and *Eurotium cristatum* particle metabolites (ECP) sifted from ECT after eCS-induced injury in mice. Sixty C57BL/6 mice were randomly divided into a blank control group, an eCS model group, an eCS + 600 mg/kg ECP treatment group, an eCS + 600 mg/kg ECT treatment group, an eCS + 600 mg/kg ECP prevention group, and an eCS + 600 mg/kg ECT prevention group. The results show that ECP and ECT significantly reduced the eCS-induced oxidative stress and inflammation and improved histopathological changes in the lungs in mice with eCS-induced liver injury. Western blot analysis further revealed that ECP and ECT significantly inhibited the eCS-induced upregulation of the phosphorylation levels of the extracellular Regulated protein Kinases (ERK), c-Jun N-terminal kinase (JNK) and p38mitogen-activated protein kinases (p38MAPK) proteins, and significantly increased the eCS-induced downregulation of the expression levels of the pregnane X receptor (PXR) and aryl hydrocarbon receptor (AhR) proteins. Conclusively, these findings show that ECP and ECT have a significant repairing effect on the damage caused by eCS exposure through the MAPK and PXR/AhR signaling pathways; ECT has a better effect on preventing eCS-induced injury and is suitable as a daily healthcare drink; ECP has a better therapeutic effect after eCS-induced injury, and might be a potential therapeutic candidate for the treatment of eCS-induced injury.

## 1. Introduction

Electronic-cigarette (eC) liquids are mainly composed of tobacco extract, glycerin, propylene glycol and nicotine. eC liquids are relatively harmless to human health, have a wide range of fashionable flavors, and are becoming popular among young smokers as an alternative to traditional cigarettes [1,2]. However, after high temperature atomization, eC liquid will still produce a small amount of formaldehyde, acetaldehyde, acrolein and glyoxal and other harmful substances [3,4,5]. Related studies have shown that electronic cigarette smoke (eCS) can increase the level of malondialdehyde(MDA) in serum [6], and decrease the activity of superoxide dismutase (SOD), catalase (CAT) and glutathione-S-transferase (GST) [7], and increase the content of interleukin-1β (IL-1β), interleukin-6 (IL-6) and tumor necrosis factor alpha (TNF-α) [8]. The mitogen-activated protein kinase (MAPK) signaling pathway is involved in the regulation of cell growth and differentiation, inflammation and apoptosis through ERK, JKN and p38. Guan’s study showed that cigarette smoke exposure could activate the expression of the ERK/JNK/p38MAPK signaling pathway and induce the upregulation of TNF-α, IL-1β and IL-6 [9]. However, although electronic cigarettes contain fewer harmful components than cigarettes, glyoxal and methylglyoxal produced after high-temperature atomization can induce the release of the pro-inflammatory cytokines IL-1β and IL-6 through the extracellular signal-regulated kinase ERK1/2, P38 and nuclear factor kappa B (NF-κB) pathways [10]. Moreover, related studies have shown that electronic cigarettes with or without nicotine can activate ERK1/2 and p38 [11]. In addition, pregnane X receptor (PXR) is an upstream regulator of many metabolic enzymes and transporters. It plays a role in drug metabolism and detoxification by regulating the expression of downstream genes Cytochrome P450 3A4 (CYP3A4) [12]. Aromatic hydrocarbon receptor (AhR) is a nuclear and cytoplasmic shuttle protein, which can regulate the biochemical and toxicological reactions of environmental chemicals [13], and mainly mediates the expression regulation of the Cytochrome P450 1A1 (CYP1A1) and Cytochrome P450 1A2 (CYP1A2) genes [14,15]. Therefore, inhibiting the activation of the MAPK signaling pathway, and while promoting the activation of the PXR/AhR signaling pathways, may be helpful in reducing the harm induced by eCS, and it is of great significance for the treatment of eCS-induced injury to find an agent which could improve these pathways. 

Recent studies showed that *Eurotium cristatum* is a unique probiotic in Fu brick tea (FBT) [16], possessing anti-tumor and anti-oxidation effects and the ability to regulate intestinal microorganisms [17,18]. FBT is a reprocessed tea from primary loose dark tea (PDT) that is fermented by *Eurotium cristatum* after being pressed. The amount of *Eurotium cristatum* has been considered an important indicator of FBT quality [19]. The water extract of FBT can inhibit the MAPK and nuclear factor-erythroid 2-related factor-2 (Nrf2) signaling pathway in human keratinocytes (HaCaT) and reduce oxidative stress levels [20]. ECT is made from the primary loose dark tea (PDT) fermented by *Eurotium cristatum*, and the number of *Eurotium cristatum* on the surface of ECT is more than that of compressed FBT. On the basis of these investigations, we speculated that ECP and ECT have protective effects on eCS-induced tissue damage. However, studies on the application of ECP and ECT to eCS have not been reported. Thus, the aim of this work was to explore the ability of ECP and ECT to improve lung injury and hepatotoxicity induced by eCS in order to provide alternative anti-eCS drugs. Therefore, in this study, we established an eCS-damaged model of C57BL/6 female mice; ECP and ECT were used to prevent and treat mice, respectively, and the effects of oxidative stress and inflammatory factor levels in serum were investigated. In addition, the underlying potential mechanism was partially elucidated using Western blot analysis of the MAPK, PXR and AhR signaling pathways.

## 2. Materials and Methods

### 2.1. Preparation of ECP and ECT Extract

The primary loose dark tea (PDT) was sterilized at 121 °C for 20 min, and was inoculated with *Eurotium cristatum* after cooling. Fermentation was carried out in an incubator at 28 °C and 80% humidity. After drying at 70–80 °C for 120 min, we collected ECT and stored it at −20 °C in a refrigerator. The processed ECT was sifted through a 100-mesh sieve, and the ECP at the bottom of the sieve and the ECT on the sieve surface were collected. They were was extracted by ultrasonic extraction (40 kHz) in ultrapure water for 30 min at 100 °C and then concentrated, freeze-dried and decompressed to obtain ECT and ECP powders. The dosages of ECP and ECT depends on the body weight (600 mg/kg) of mice. The dosages of ECP and ECT were selected based on references and preliminary experiments [21].

### 2.2. Analysis of Physicochemical Components of ECP and ECT

The contents of the water extract, tea polyphenols and free amino acids were determined according to GB/T 8305-2013, GB/T 8313-2008 and GB/T8314-2013, respectively. The contents of catechin, gallic acid and caffeine were analyzed by HPLC according to the method of Huang [21]. The HPLC analysis of quercetin and kaempferol was carried out according to the method of Samanidou [22].

### 2.3. Metabolomic Analysis of ECP and ECT

#### 2.3.1. Sample Preparation and Extraction

The 150 mg ECP and ECT samples were placed in a 2 mL grinding thickening tube, and 1 mL of extraction solution (methanol/water = 7:3, pre-cooled at −20 °C), and two small steel balls were added to the test tube, and grinded in a tissue grinder (50 Hz for 5 min). The samples were placed at 4 °C, vortexed once every 10 min for a total of 3 times, and then 4 °C overnight. The next day, they were vortexed and centrifuged at 13,000× *g* for 10 min at 4 °C. After centrifugation, 800 μL of the supernatant was taken and passed through a 0.22 μm filter membrane, and the filtered samples were placed in a sample bottle for LC-MS analysis. 

#### 2.3.2. UPLC-MRM Analysis

In this experiment, a Waters ACQUITY UPLC I-Class (waters, USA) tandem QTRAP6500 Plus high-sensitivity mass spectrometer (SCIEX, USA) was used for the separation and quantitative detection of metabolites. The chromatographic column was an ACQUITY UPLC HSS T3 column (100 × 2.1 mm, 1.8 μm, Waters). The mobile phase was an aqueous solution containing 0.1% formic acid (liquid A) and 100% acetonitrile (liquid B) containing 0.1% formic acid. Elution B was carried out with the following gradient: 0~2 min, 5%; 2~22 min, 5~95%; 22~27 min, 95%; 27~30 min, 5%. An injection volume of 8 μL, solvent flow rate of 0.3 mL/min, and column temperature of 40 °C were used. For the QTRAP 6500 Plus system with the EST Turbo ion spray interface, the ion source parameters were set as follows: the ion source temperature was set to 500 °C; the ion spray voltage (IS) was set to 4500 V (positive mode)/−4500 V (negative mode), ion source gas I (GS1), gas II (GS2) and air curtain gas (CUR) were set to 40, 40 and 20 psi, respectively. The MRM detection window was set to 120 s and the target scanning time was set to 0.5 s. 

### 2.4. eC Liquid Preparation

eCS liquid was provided by the China Tobacco Hunan Industry Co., Ltd. (Changsha, China). It consists of 95% aerosol and 5% cigarette essential oil. The atomizer consists of propylene glycol (PG), vegetable glycerin (VG), water and nicotine. Tobacco essential oil is extracted from tobacco by supercritical carbon dioxide. Its main components are nicotine, anthracene-D10, phenanthrene-D10, cyclohexene, glycerin and other substances.

### 2.5. Chemicals

Analytical kits for SOD, GSH-Px and MDA were purchased from the Nanjing Jiancheng Bioengineering Institute (Nanjing, China). ELISA kits for the analysis of tumor necrosis factor alpha (TNF-α), interleukin-6 (IL-6), IL-8, and IL-1β were purchased from Wuhan Hualianke Biotechnology Co. Ltd. (Wuhan, China). Extracellular signal-regulated kinases (ERK), phospho-ERK (p-ERK), c-Jun N-terminal kinase (JNK), phosphor-JNK (p-JNK), p38 and phosphor-p38 (p-p38) were purchased from Cell Signaling Technology (Danvers, MA, USA). Monoclonal antibodies against PXR (ab192579), AhR (ab84833), and GAPDH (ab181602) were purchased from Abcam (Cambridge, UK). All other chemicals and reagents were of analytical grade.

### 2.6. Animals and Experimental Design

C57BL/6SPF female mice (15 ± 1 g) were purchased from Changsha Slakejingda Experimental Animal Co., Ltd., and with the experimental animal production license number: SCXK (Hunan) 2016-0002. Approved by the Animal Test Committee of Hunan Agricultural University, the mice were raised in the Animal Experimental Center of the Tea Research Institute of Hunan Agricultural University. The temperature of the feeding environment was (25 ± 1) °C, the humidity was 40–70%, and the light time was 12 h day and night. After two weeks of adaptive feeding, 60 mice were randomly divided into 6 groups: (1) blank control group; (2) eCS exposure model group; (3) eCS + ECP treatment group; (4) eCS + ECT treatment group; (5) eCS + ECP prevention group; (6) eCS + ECT prevention group. Mice in the blank control group and the eCS group received an equal volume of distilled water by gavage every day for 12 weeks; meanwhile, mice in the eCS group were exposed to eCS from the 5th week. In the first 8 weeks, the mice in the treatment group received an equal volume of distilled water by gavage and were exposed to eCS. From the 9th week to the 12th week, the mice in the treatment group were no longer exposed to eCS and were fed with ECT and ECP of 600 mg/kg each of ECT and ECP, respectively. In the first 4 weeks, the mice in the prevention group received an equal volume of distilled water by gavage. From the 5th week to the 12th week, the mice in the prevention group were fed with ECT and ECP at the concentration of 600 mg/kg immediately after daily eCS exposure (Figure 1). The mice in the eCS exposure group were exposed to a self-made passive smoking box on the first day for 12 min, increased by 5 min every day until 60 min was reached. The self-made passive smoking device for mice (0.9 m × 0.6 m × 0.5 m plastic box) evenly distributes 20 vents with a radius of 2 cm around and at the top. Vacuum diaphragm pumps were purchased from Kamoer, KVP15-KL-1 (Shanghai, China). The rubber tube at the air inlet was connected to the electronic cigarette smoker, and the rubber tube at the air outlet extended into the box to release smoke. For the smoking frequency, GB/T16450-2004 was referred to in order to establish the smoke damage model. 

### 2.7. Collection of Serum and Lung and Liver Tissues in Mice

The mice fasted for 12 h before death and were anesthetized with pentobarbital sodium. The serum was collected from the eyeballs, left at room temperature for 1 h, centrifuged for 10 min at 4 °C for 2500 r/min, and stored at −80 °C. The lung and liver tissues were removed and washed 3 times in normal saline at 4 °C. The surface water and blood stains were dried using sterile filter paper. The right lung tissues were placed in formalin to prepare pathological sections; the remaining lung and liver tissues were stored at −80 °C for follow-up testing.

### 2.8. Histological Evaluation

The right lung tissue was fixed in formalin for 3 days, embedded in paraffin, and stained with hematoxylin-eosin solution. Optical microscopy was used to assess the morphological changes in the lung tissue.

### 2.9. Biochemical Analysis

The oxidative stress index in the serum of mice was evaluated by measuring the activity of GSH-Px and SOD and the content of MDA in the serum using a biochemical kit. The serum levels of TNF-α, IL-8, IL-6 and IL-1β were determined by ELISA kits to evaluate the inflammatory reaction. All the tests were carried out according to the instructions of the reagent.

### 2.10. Western Blot Analysis

The lung and liver tissue proteins were extracted using a total protein kit (Solarbio, Beijing, China), and tissue protein concentration was quantified to 1 μg/μL using a BCA kit (Solarbio, Beijing, China). An amount of 20 μg of denatured proteins was separated in 10% polyacrylamide gel electrophoresis (80 V, 25 min, 120 V, 60 min); then, the target protein was transferred to PVDF membranes (300 mA, 90 min). Afterwards, the membranes were sealed in TBS-Tween (TBST) containing 5% skim milk powder for 1 h. The closed membranes were washed in a shaker with TBST 3 times, for 10 min each time. The membranes were incubated overnight in a primary antibody (used at a dilution of 1:10,000) at 4 °C, and the membranes were washed three times with TBST and incubated with a secondary antibody (used at a dilution of 1:10,000) at room temperature for 1.5 h. After the membranes were washed with TBST three times, the membranes were added to a chemiluminescent mixture, and were exposed to a chemiluminescence imager to display the target protein. The image was transformed by imaging software, the gray ratio of the target strip to the GAPDH internal reference strip was evaluated and the experiment was repeated 3 times.

### 2.11. Statistical Analysis

The metabolites were identified and quantitatively analyzed using the MRM quantitative software MultiQuant (AB Sciex, Framingham, MA, USA), combined with a widely targeted metabolic standard database (BGI-WideTarget-Library) independently established by the BGI (Shenzhen, China). Bioinformatics analysis was carried out using the OmicStudio tool available at https://www.omicstudio.cn/tool (accessed on 11 February 2022). The data were analyzed and processed by IBM SPSS Statistics 22.0 software, and the significant differences among groups were analyzed by single-factor analysis of variance (one-way ANOVA). According to the homogeneity of variance, the LSD method and Tamhane method were adopted. All experimental data were expressed as the mean ± SD, and plotted with GraphPad Prism 7, Adobe Acrobat DC and Photoshop CS6. Statistical significance was set at *p* < 0.05 or *p* < 0.01. 

## 3. Results

### 3.1. Analysis of Physicochemical Components of ECT and ECP

It can be seen from Table 1 that, except for ECG, the contents of the other components in ECT were much higher than those in ECP, especial for tea polyphenols, caffeine and gallic acid. There is no specific standard for the analysis of physicochemical components of ECP, so the current HPLC detection method was only able to detect a small amount of ECP content. However, the similar content of the two water extracts indicates that the ECP extract may contain undetected chemical substances, but further detection and research are needed. 

### 3.2. Metabolomic Analysis of ECP and ECT

#### 3.2.1. Principal Component Analysis (PCA) and Partial Least Squares-Discriminant Analysis (PLS-DA) of ECT and ECP

In the unsupervised PCA score chart (Figure 2A), the first and second principal components explained 89.31% and 2.32% of the variation, respectively. In the supervised PLS-DA score chart (Figure 2B), the first and second principal components explained 90.78% and 1.85% of the variation, respectively. There are obvious differences in the distribution of Figure 2A,B between the ECT and ECP samples, indicating that there are significant differences between them. Emergency aggregation appeared in the ECT and ECP samples, which confirmed the stability and reproducibility of the analysis method. The interpretation rate R2 and predictive power Q2 of PLS-DA both exceeded 0.9. Cross-validation of 100 permutation tests showed that the R2 and Q2 intercepts were 0.97 and −0.75, respectively, showing that the Q2 intercept was less than 0, which indicates that the PLS-DA model is reliable (Figure 2C).

#### 3.2.2. Heatmap Analysis of the Differential Metabolites of ECT and ECP

Heatmaps were drawn to visualize the difference between the 212 compounds in ECT and ECP, with each column representing a sample and each row representing a metabolite (Figure 3). Red indicates that the metabolite content is above the sample average, while blue indicates that the metabolite content does not reach the average level. It can be seen from the heatmaps that, comparing ECT and ECP, ECT contains more phenols and their derivatives, phenolic acids, glycosides, flavonoids, lignans, terpenes and other substances, while ECP contains more amino acids and their derivatives, organic acids, alkaloids, carbohydrates and other substances, and the amounts of esters and coumarin and its derivatives are equal. 

#### 3.2.3. Boxplot Analysis of the Differential Metabolites of ECT and ECP

Based on functional pharmacology, 39 bioactive metabolites with the highest content of ECT and ECP were selected from the differential metabolites’ heatmaps to draw the boxplots, and the content difference was analyzed in detail. There were 18 metabolites with a higher content in ECT and 21 metabolites with a higher content in ECP. It can be seen from Figure 4 that the contents of C, EC, EGC, ECG and GCG in ECT are higher than those in ECP, but the difference is not very significant, being only 1–5 times higher than those of ECP. The contents of eight flavonoid metabolites such as morin, vitexin, tricetin, licochalcone B, eriodictyol, dihydrokaempferol, hesperetin and naringenin7-O-glucoside, in ECT were higher than those in ECP. Apart from that, the content of vitexin in ECT, which was about 5 times higher than that in ECP, the content of seven other flavonoids was more than 10 times higher than that in ECP. The two glycosides oroxin A and sophoroside in ECT were two and three times higher than those in ECP, respectively. The three organic acids gallic acid, 5-acetylsalicylic acid and isochlorogenic acid B in ECT were two, three and five times higher than those in ECP, respectively. 

The contents of D-mannose, deoxyglucose, L-fucose, mannitol and lactose in ECP were higher than those in ECT. Except for mannose, the other four sugars were more than 10 times higher than ECT. The contents of L-hydroxyproline, L-leucine, L-tyrosine, L-phenylalanine and their derivatives in ECP were higher than those in ECT, and the lowest difference was more than 30 times. The contents of five flavonoid metabolites in ECP, namely, isobavachin, isobavachalcone, licochalcone A, 7,8-dihydroxyflavone and dihydromyricetin, were higher than those in ECT. Apart from the content of dihydromyricetin, which was only slightly higher than that in ECT, the contents of the other four flavonoids were more than 10 times higher than that in ECT. The contents of heobromine, theophylline, caffeine and betaine in ECP were higher than those in ECT. Apart from the content of betaine, which was nearly 40 times higher than that in ECT, the contents of the other three alkaloids were only about 2 times those in ECT. The content of protocatechuic acid in ECP was more than 10 times higher than that in ECT. It is worth noting that the content of three other metabolites in ECP, namely, steviolbioside, kahweol and ginkgolide K, was more than 50 times higher than that in ECT.

### 3.3. Pathological Changes in Lung Tissue Sections in Mice

The pathological section of the lung tissue of mice is shown in Figure 5. Compared with the blank control group, the eCS group showed an irregular arrangement of tracheal cilia, thickening of alveolar wall, rupture of the alveolar septum, dilation of the cavity, infiltration of inflammatory cells and obvious blood osmosis. Compared with the eCS exposure model group, the lung tissue trachea of mice in the prevention group and treatment group of ECP and ECT was more regular, the alveolar wall became thinner, the degree of alveolar septum injury was attenuated, the wall expansion became smaller, the inflammatory cell infiltration decreased and there was no obvious blood osmosis. ECP and ECT intragastric intervention improved lung tissue injury, the alveolar tissue morphology nearly returned to the normal level, and the treatment group had a better anti-inflammatory potential than the preventive treatment group.

### 3.4. Changes in Serum Oxidative Stress Indexes and Inflammatory Factors in Mice

Whether ECT and ECP can alleviate oxidative stress and the inflammatory response caused by eCS exposure is the key to whether ECT and ECP have preventive and therapeutic effects. According to the results in Figure 6, Compared with the blank control group, the contents of TNF-α, IL-8, IL-6, IL-1β and MDA in the serum of the eCS model group were significantly increased (*p <* 0.01), while the activities of GSH-Px and SOD were significantly decreased (*p <* 0.01). Compared with the eCS model group, the serum oxidative stress index of mice treated with ECP and ECT were improved. The ECP and ECT treatment groups had a significantly reduced level of MDA in the serum and increased activities of GSH and SOD (*p <* 0.05). The ECP and ECT prevention groups also had a reduced level of MDA in the serum, and increased activities of GSH and SOD, and there were significant differences in the ECT prevention group (*p <* 0.05). Compared with the eCS model group, the inflammatory factors in serum of mice treated with ECP and ECT were improved. The contents of TNF-α, IL-6 and IL-8 in the ECP prevention group were significantly decreased (*p <* 0.05), but the contents of TNF-α, IL-1β, IL-6 and IL-8 in the ECP and ECT treatment groups and the ECT prevention group were significantly decreased (*p <* 0.01). 

### 3.5. Changes in Relative Expression of ERK, JNK and p38 MAPK Phosphorylated Proteins in Mousee Lung

The phosphorylation of the MAPK pathway is one of the key characteristics of oxidative stress and inflammation caused by eCS. According to the results in Figure 7, compared with the blank control group, the phosphorylation levels of the ERK, JNK and p38 proteins in the lungs of the eCS model group were significantly increased (*p* < 0.01). Compared with the eCS model group, the phosphorylation level of the ERK and JNK proteins in the ECP and ECT treatment groups decreased significantly (*p* < 0.01). The phosphorylation level of the p38 protein in the ECP treatment group decreased significantly (*p* < 0.05), and the phosphorylation level of the p38 protein in the ECT treatment group decreased, but there was no significant difference (*p* > 0.05). The phosphorylation levels of the JNK protein significantly decreased in the ECP prevention group (*p* < 0.05), and the phosphorylation levels of the ERK and P38 protein were decreased but there was no significant difference (*p* > 0.05). The phosphorylation levels of ERK and P38 significantly decreased in the ECT prevention group (*p* < 0.05), while the phosphorylation levels of JNK decreased, but there was no significant difference (*p* > 0.05).

### 3.6. Changes in Relative Expression of PXR and AhR Proteins in Liver Tissue of Mice

To assess the preventive and therapeutic ability of ECT and ECP against eCS-induced liver damage, the PXR/AhR signaling pathway was investigated by Western blot. According to the results in Figure 8, the protein expression of PXR in the liver of the eCS model group was lower than those of the normal control group (*p* > 0.05); the protein expression of AhR in the liver of the eCS model group was significantly lower than those of the normal control group (*p* < 0.01). Compared with the eCS model group, the protein expression of PXR and AhR in the ECP and ECT treatment groups significantly increased (*p* < 0.05), and the protein expression of PXR and AhR in the ECP prevention group increased, but there was no significant difference (*p* > 0.05). The protein expression of PXR and AhR in the ECT prevention group significantly increased (*p* < 0.05). 

## 4. Discussion

The results show that both ECP and ECT can have antioxidant and anti-inflammatory effects in the lung through the MAPK pathway and detoxification roles in liver through the PXR/AhR pathway to decrease eCS-induced injury. Oxidative stress induced by eCS plays an important role in the process of causing lung and liver injury. ECP and ECT could improve oxidative stress and increase the activity of antioxidant enzymes in serum, which may play an antioxidant role in eCS-induced oxidative damage by inhibiting MDA and promoting the production of GSH and SOD. Overall, the antioxidant capacity of the treatment group was better than that of the prevention group, which may be due to the mice in the treatment group not being exposed to eCS in the third month. It is worth noting that the therapeutic effect of ECP is better than that of ECT, while the preventive effect of ECT is better than that of ECP. The other feature of damage caused by eCS is inflammation, and oxidative stress can also amplify the degree of inflammatory damage [23,24]. eCS and its atomized harmful substances can induce the release of pro-inflammatory cytokines, such as TNF-α, IL-1β, IL-6 and IL-8, thus aggravating inflammation and apoptosis [9,25]. The results of this study show that the levels of the TNF-α, IL-1β, IL-6 and IL-8 pro-inflammatory factors in mice exposed to eCS increased significantly, but both ECP and ECT reversed this situation; the trend of the anti-inflammatory effect was that the therapeutic effect was better than the preventive effect, and the therapeutic effect of ECP was better than that of ECT, while the preventive effect of ECT was better than that of ECP. This result is also consistent with the lung histopathological observation.

To further explore the mechanism of oxidative stress and anti-inflammation induced by ECP and ECT on eCS exposure through the MAPK pathway, studies have shown that cigarette smoke is a potential ERK trigger that can induce inflammation and oxidative stress by activating ERK signaling cascades in the lungs [26]. In chronic obstructive pulmonary disease (COPD), activation of ERK produces pro-inflammatory cytokines, such as TNF-α, IL-1β, IL-6 and IL-8 [27,28]. JNK and p38 are also two important signaling pathways in the MAPK family. Phosphorylated by signaling factors, they can act on a variety of downstream transcription factors and produce stress responses such as inflammation [29,30]. eCS can activate the p38 MAPK pathway and promote inflammation, which may be caused by acrolein produced by the heating of glycerol in eCS [31,32]. Glyoxal and methyl glyoxal in electronic cigarette smoke induce the expression of proinflammatory cytokines in human nasal epithelial cells throughERK1/2 and p38MAPK [10]. In addition, some studies have shown that nicotine can cause oxidative stress and activate p38 MAPK, but it is not the only factor that activates the MAPK pathway [11,33]. In our experimental study, long-term exposure to eCS significantly increased the level of ERK, JNK and p38 MAPK phosphorylation in the lung tissue of mice, and the treatment group showed a good therapeutic effect in reducing the level of MAPK phosphorylation, except that the ECT treatment group could not effectively reduce the level of p38 phosphorylation. ECT treatment also decreased the phosphorylation levels of ERK and p38 MAPK in the prevention group, but the effect was not as good as that in the treatment group, which may be due to the fact that the phosphorylation ability of MPAK caused by daily exposure to eCS was better than the repair ability of ECP and ECT, and the effect of ECP prevention group on reducing the phosphorylation level was less obvious. It is worth noting that ECP treatment tends to normalize the level of MAPK phosphorylation caused by eCS. In addition to the immune repair ability of the body, according to the previous metabolomics data, ECP’s excellent therapeutic effect may be related to its high content of amino acids and their derivatives, organic acids, alkaloids and sugars. In addition, other high levels of active substances in ECP also play an important role in the MAPK pathway, such as protocatechuic acid, ginkgolide K and kahweol. Protocatechuic acid can inhibit the inflammation induced by lipopolysaccharide in primary keratinocytes by reducing the activation of the Toll-like receptor (TLR4)/NF-κB signaling pathway and of the JNK and p38-MAPK pathways [34]. Ginkgolide K can reduce oxygen-glucose-induced SH-SY5Y cell injury by inhibiting the p38 and JNK signaling pathways [35]. Kahweol may inhibit the phosphorylation of JNK1/2 and p38 MAPK and the activation of NF- κ B to reduce the expression of phorbol 12-myristate 13-acetate (PMA)-enhanced matrix metalloproteinase-9 (MMP-9) to promote an anti-tumor effect [36]. In addition, studies have shown that the extracellular polysaccharide secreted by *Eurotium cristatum*, which is composed of mannose, glucose and galactose, has a strong immunomodulatory ability. The polysaccharide controls the activation of phagocytes, and indirectly kills pathogens by improving the levels of TNF-α, IL-1β, IL-6, IFN-γ and NO, and it has shown effective immunomodulatory activity [37,38]. It is also associated with the ability of the cell surface receptor TLR4 to bind, which mediates the activation of the signal-regulated protein kinases ERK, JNK and p38 MAPK, thus inducing the expression of immune factor genes [39]. In addition, in Eurotium cristatum, an unreported allyl benzaldehyde derivative, cristaldehyde A, and an unreported quinone derivative, cristaquinone A, have been found, and its five known compounds also have significant anti-inflammatory and free radical scavenging activity [40]. This is consistent with the results of previous oxidative stress and inflammatory factor level tests, which suggests that the protective mechanism of ECP and ECT against eCS-induced lung injury may be at least partly attributed to the enhancement of antioxidant and anti-inflammatory defense systems by inhibiting the MAPK signaling pathway.

Pregnane X receptor (PXR) and aromatic hydrocarbon receptor (AhR) are mainly expressed in the liver and participate in regulating drug metabolism enzymes in the liver. PXR is a primary chemical sensor, and it is known that it can induce the expression of the CYP3A enzyme to regulate substance metabolism and detoxification function. AhR is a ligand-activated transcription factor that mediates downstream regulatory genes of the CYP1A enzyme to help decompose environmental toxins [41,42]. However, some studies have shown that there is crosstalk between them in regulating the expression of cytochrome P450 family genes in the liver [43]. In this study, eCS significantly decreased the levels of PXR and AhR, and this situation was reversed after intragastric administration of ECP and ECT; however, the effect of the treatment group was still better than that of the prevention group. It is worth noting that the ECP treatment group tended to normalize the decline in PXR levels caused by eCS exposure, showing an excellent detoxification effect. This phenomenon also corresponds to the excellent regulation ability of the ECP group in the MAPK pathway, indicating that ECP is more suitable as a therapeutic drug for eCS exposure injury at the same concentration as ECT. However, surprisingly, like the previous indicators of oxidative stress, inflammatory factors and the MAPK pathway, the ECT prevention group was better than the ECP prevention group in reversing the decrease in PXR and AhR induced by eCS exposure. Therefore, we speculate that ECT without intentional removal of ECP contains not only phenols and their derivatives, phenolic acids, glycosides, flavonoids, lignans, terpenes and other active substances in tea, but also amino acids and their derivatives, organic acids, alkaloids and carbohydrates in ECP, which activate various immune factors and trend factors and have a more comprehensive immunomodulatory function. It can enhance the resistance of mice to electronic cigarette smoke and reduce the degree of damage caused by daily eCS exposure. In addition to detoxification, drug metabolic enzyme reactions can lead to biological activation, causing inflammation, leading to tissue damage [44]. Some studies have shown that PXR is the main receptor for exogenous substances, and the inflammatory environment causes the activation of the NF-κB transcription factor, which inhibits the activity and function of the PXR protein, leading to the decline in the down-stream expressed CYP3A enzyme, and finally down-regulating the drug metabolism ability [45]. Zhang [46] showed that activation of PXR can inhibit NF-κB activity to reduce inflammation; there is a feedback regulation in this pathway. Related studies have reported that the AhR binding site is very close to or overlaps with the NF-κB binding site, which suggests that AhR and NF-κB may have potential intermodulation and crosstalk effects [47]. Experimental studies in animal models of various inflammatory diseases have demonstrated that activation of AhR can participate in a variety of cellular functions such as the regulation of the immune system and the suppression of the inflammatory response [48,49,50]. The experimental results of the PXR/AhR signaling pathway are indeed consistent with the reported results and a negative trend was observed between PXR/AhR and the level of inflammation (Figure 9). However, how the nuclear receptor PXR/AhR and the NF-κB pathway transcriptionally regulate the inflammation induced by eCS exposure, as well as how ECP and ECT reduce the inflammation caused by eCS exposure in this signaling pathway, remains unclear, and we need further experiments to elucidate this. 

ECP and ECT can effectively ameliorate the lung injury and hepatotoxicity caused by eCS. In addition, the good therapeutic effect of ECP and the good preventive effect of ECT can be considered for synergistic use against lung tissue damage caused by eCS, but some studies have shown that the drug–drug interaction may lead to adverse drug reactions, or even synergistic toxicity [51]. Therefore, whether the combination of ECP and ECT will cause hepatotoxicity or not needs to be verified through further experiments.

## 5. Conclusions

ECP and ECT down-regulated the phosphorylation levels of the ERK, JNK and p38 proteins in mouse lungs induced by eCS and reversed the decrease in the PXR and AhR protein levels in mouse livers induced by eCS. In conclusion, ECP and ECT reduced the oxidative stress and inflammatory injury caused by eCS through the MAPK pathway, and the hepatotoxicity caused by eCS through the PXR/AhR pathway. Therefore, ECT may be used in a healthy drink to reduce the risk of eCS-induced injury, and ECP might be a potential therapeutic candidate for the treatment of eCS-induced injury.

## Figures and Tables

**Figure 1 nutrients-14-02843-f001:**
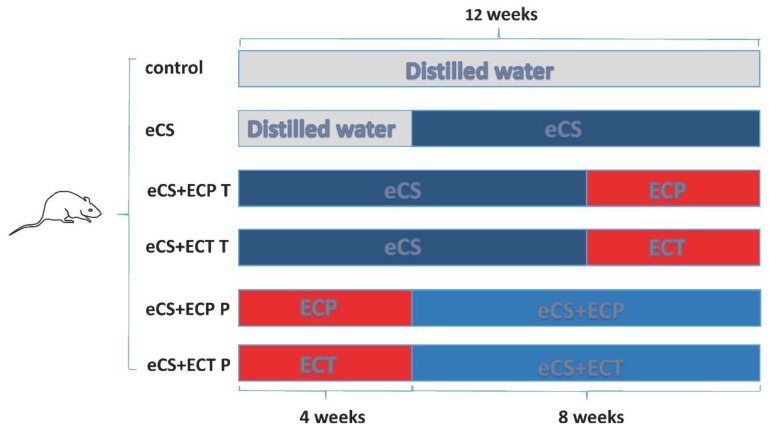
The schematic diagram of experimental groupings.

**Figure 2 nutrients-14-02843-f002:**
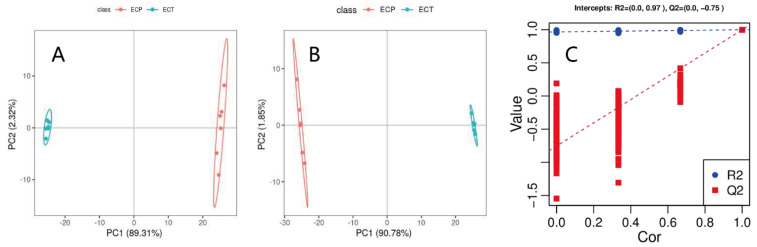
(**A**) PCA score chart of ECT and ECP; (**B**) PLS−DA score chart of ECT and ECP; and (**C**) response sequencing test diagram of PLS−DA analysis model.

**Figure 3 nutrients-14-02843-f003:**
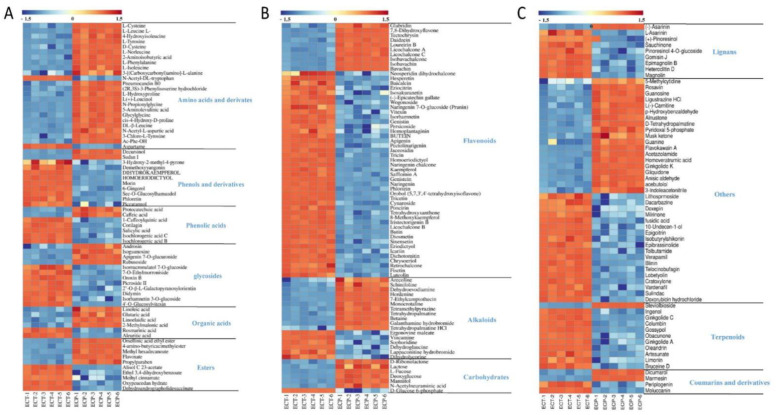
(**A**) Heat map of amino acids and derivates, phenols and derivatives, phenolic acids, glycosides, organic acids and esters in ECT and ECP; (**B**) Heat map of flavonoids, alkaloids and carbohydrates in ECT and ECP; (**C**) Heat map of lignans, coumarins and derivatives, terpenoids and other differential metabolites in ECT and ECP.

**Figure 4 nutrients-14-02843-f004:**
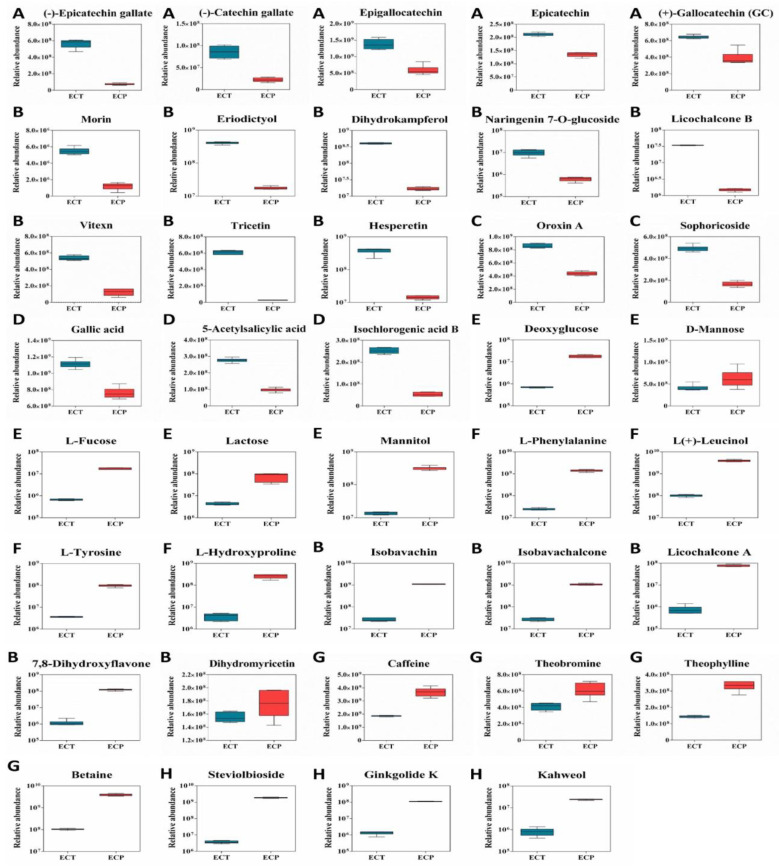
39 differential metabolites box diagram of ECT and ECP. (**A**): Catechin; (**B**): flavonoids; (**C**): flavonoids; (**D**): flavonoids; (**E**): flavonoids; (**F**): amino acids; (**G**): alkaloids; (**H**): other metabolites.

**Figure 5 nutrients-14-02843-f005:**
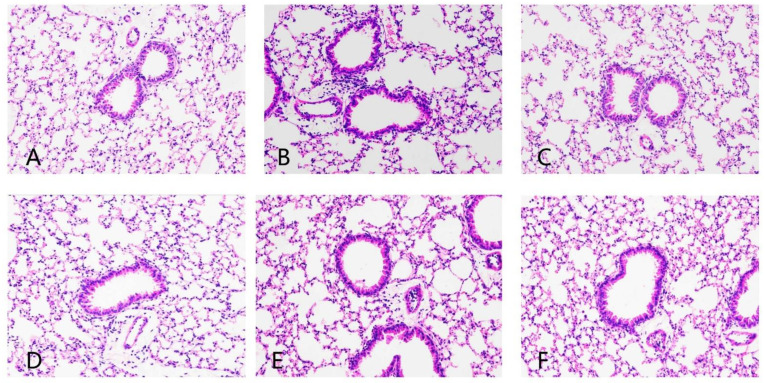
The effects of ECP and ECT on lung histopathology in mice with eCS exposure-induced lung injury (200×). (**A**): Control; (**B**): eCS; (**C**): eCS + ECP T; (**D**): eCS + ECT T; (**E**): eCS + ECP P; (**F**): eCS + ECT P.

**Figure 6 nutrients-14-02843-f006:**
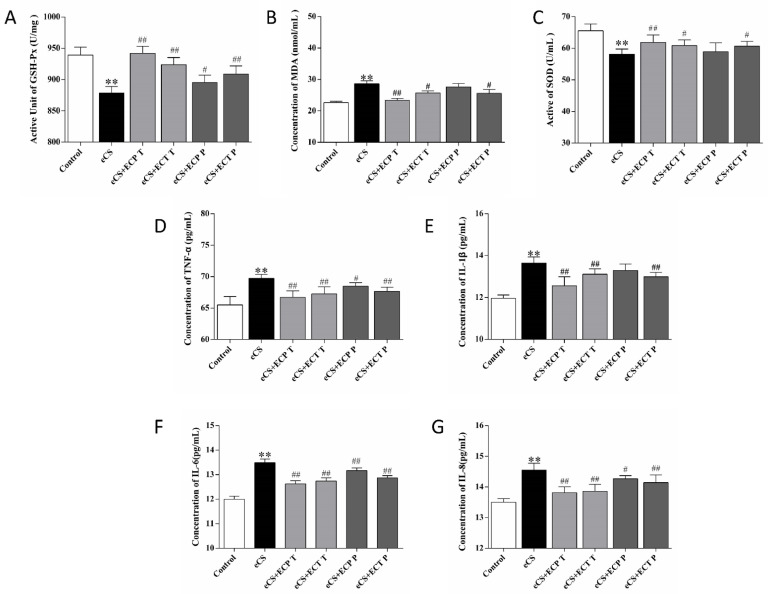
The effects of ECP and ECT on GSH-Px and SOD activities and the expression of MDA, TNF-α, IL-6, IL-8, and IL-1β levels in serum of mice exposed to eCS. (**A**) Active unit of GSH-Px. (**B**) Activity of SOD. (**C**) Concentration of MDA. (**D**) Concentration of TNF-α. (**E**) Concentration of IL-1β. (**F**) Concentration of IL-6. (**G**) Concentration of IL-8. The measures represent mean ± SD. **: compared with the blank control group, *p <* 0.01; #: compared with the eCS model group, *p* < 0.05; ##: compared with the eCS model group, *p* < 0.01.

**Figure 7 nutrients-14-02843-f007:**
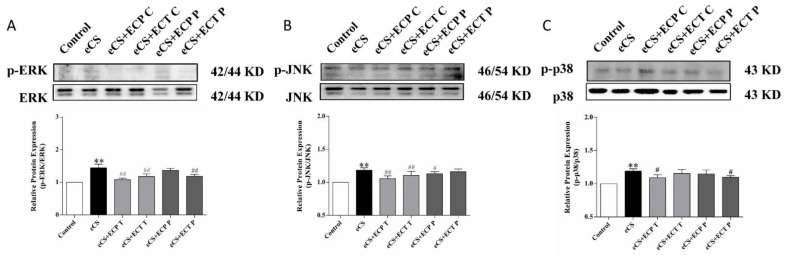
The effects of ECP and ECT on the expression of the p38, p-p38, JNK, p-JNK, ERK and p-ERK proteins in the lung of mice exposed to eCS. Western blot analysis for the expression of phosphorylated (**A**) p-ERK and total ERK, (**B**) p-JNK and total JNK, (**C**) p-p38 and total p38 in lung tissues of mice induced by eCS. The measures represent mean ± SD. **: compared with the blank control group, *p <* 0.01; #: compared with the eCS model group, *p* < 0.05; ##: compared with the eCS model group, *p* < 0.01.

**Figure 8 nutrients-14-02843-f008:**
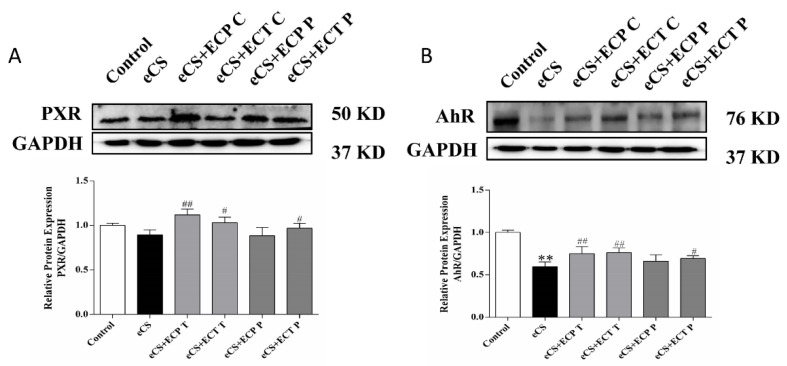
The effects of ECP and ECT on the expression of the PXR and AhR protein in the liver of mice exposed to eCS. Western blot analysis for the expression of (**A**) PXR and (**B**) AhR in liver tissues of mice induced by eCS. The measures represent mean ± SD. **: compared with the blank control group, *p <* 0.01; #: compared with the eCS model group, *p* < 0.05; ##: compared with the eCS model group, *p* < 0.01.

**Figure 9 nutrients-14-02843-f009:**
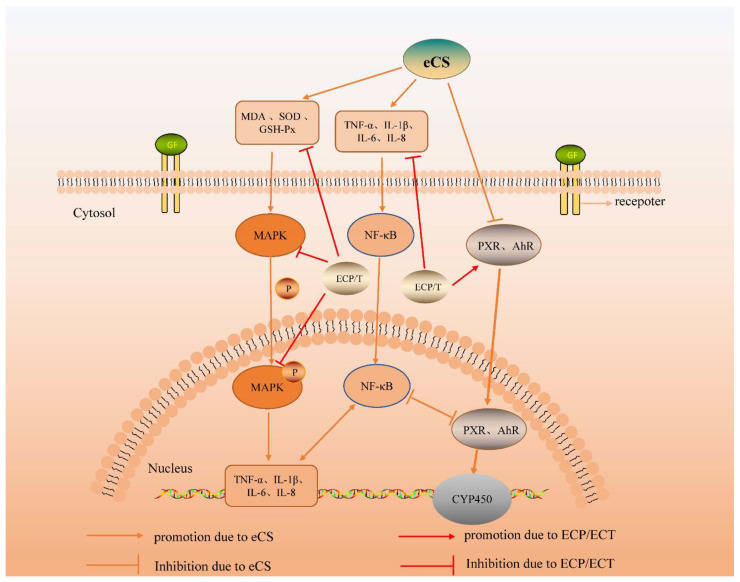
Effect of ECP and ECT on MAPK signaling pathways in eCS-induced lung injury and PXR/AhR signaling pathway in eCS-induced liver injury in mice.

**Table 1 nutrients-14-02843-t001:** Change of physicochemical components of ECT and ECP.

Physicochemical Components	ECT	ECP
Tea polyphenols (%)	14.80 ± 0.22	3.40 ± 0.21 *
Water extract (%)	41.55 ± 0.37	39.13 ± 1.71
Free amino acid (%)	3.02 ± 0.02	2.62 ± 0.01 *
Gallic acid (mg/g)	13.23 ± 0.66	2.14 ± 0.20 *
Theobromine (mg/g)	1.82 ± 0.06	1.42 ± 0.32
Theophylline (mg/g)	0.41 ± 0.12	0.08 ± 0.02 *
Caffeine (mg/g)	39.40 ± 1.25	5.65 ± 0.208 *
EGC (mg/g)	5.12 ± 0.32	2.74 ± 0.10 *
C (mg/g)	6.41 ± 0.15	0.17 ± 0.01 *
EC (mg/g)	3.12 ± 0.05	0.08 ± 0.01 *
EGCG (mg/g)	9.86 ± 0.17	5.39 ± 0.57 *
GCG (mg/g)	7.87 ± 0.23	2.27 ± 0.34 *
ECG (mg/g)	1.68 ± 0.11	4.89 ± 0.43 *
Quercetin (mg/g)	0.82 ± 0.08	0.18 ± 0.09 *
Kaempferol (mg/g)	1.49 ± 0.217	0.20 ± 0.04 *

(+)-catechin (C), (−)-epicatechin (EC), (−)-gallocatechin gallate (GCG), (−)-epigallocatechin (EGC), (−)-epicatechin gallate (ECG), (−)-epigallocatechin gallate (EGCG). *: compared with the ECT group, *p* < 0.05.

## Data Availability

The data used to support the results of this study are available from the corresponding authors.

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
