# Peer review of "Protective Effect of *Eurotium cristatum* Fermented Loose Dark Tea and *Eurotium cristatum* Particle on MAPK and PXR/AhR Signaling Pathways Induced by Electronic Cigarette Exposure in Mice"

_nutrients, 2022, doi:10.3390/nu14142843_

Round 1

Reviewer 1 Report

Authors should give more explanation on figure 2, 3 legend.

The data from figure 6, the p-erk, p-JNK and p-p38 induced by eCS were not significant. 

Fgiure 7 A, The decrease of PXR on eCS is not significant. 

all the "p" value should italic. 

Reviewer 2 Report

The manuscript is nice and it shows how herbal treatment can improve lung damage due to ecigaretts. 

Comments:

1. To treat the lung injury in current scenario a number of NSAIds are being used how ever drug drug interactions are a major highlight in todays world. However AI has showed Machine learning liver-injuring drug interactions with non-steroidal anti-inflammatory drugs (NSAIDs) from a retrospective electronic health record (EHR) cohortA Datta, NR Flynn, DA Barnette, KF Woeltje, GP Miller… - PLoS computational biology, 2021) how this drug interactions occur and to prevent this there is huge importance of herbal treatment. Authors should include this paragraph with the citation cited above to show the importance of the study.

2. Fig 3 the writing on the graph is not visible please increase the size of the writeup

3. Have the authors measured any fibrotic markers ?

6. IN figure 6 the western blots are not at all clear and where are control house keeping genes for those gene expression

Reviewer 3 Report

In this paper, the authors investigated the preventive and treatment effects of Eurotium Cristatum tea and particle on the oxidative and inflammatory status of electronic cigarette exposed rats. Unlucky, the quality of the most figure is dramatically low and do not permit to Reviewer to understand them in an adequate way.

The Authors should resubmit the paper after increasing its quality.

Anyway, some considerations arose.

1)      Provide a scheme of the experimental design (a division of groups, time of exposure and supplementation…).

2)      The quality of English is poor. Check it by mother-tongue

3)      Edit the manuscript following the instructions

4)      Perform statistical analysis in table 1.

5)      The authors should explain how they have chosen the ECP and ECT concentrations to fed rats

6)      Include the aim of the study in the introduction section

7)      Explain the ECP preparation

Round 2

Reviewer 2 Report

Authors have meet the reviewers comment nicely

Author Response

请参阅附件

Reviewer 3 Report

After the first round and my suggestions to include a scheme of the experimental design and improve the quality of the figures, the paper still continues to miss this part. In particular, regarding point 1, the authors have provided only a description. Moreover, high-quality figures should be included in the manuscript not as supplementary materials, which usually contains additional data.
